# Pipeline Leak Detection and Estimation Using Fuzzy PID Observer

Raheleh Jafari [1], Sina Razvarz [2], Cristóbal Vargas-Jarillo [2], Alexander Gegov [3,4,*] and Farzad Arabikhan [3]

1   School of Design, University of Leeds, Leeds LS2 9JT, UK; r.jafari@leeds.ac.uk
2   Departamento de Control Automatico, CINVESTAV-IPN (National Polytechnic Institute), Mexico City 07360, Mexico; srazvarz@ctrl.cinvestav.mx (S.R.); cvargas@ctrl.cinvestav.mx (C.V.-J.)
3   School of Computing, University of Portsmouth, Portsmouth PO13HE, UK; farzad.arabikhan@port.ac.uk
4   English Language Faculty of Engineering, Technical University of Sofia, 1756 Sofia, Bulgaria
*   Correspondence: alexander.gegov@port.ac.uk

**Abstract:** A pipe is a ubiquitous product in the industries that is used to convey liquids, gases, or solids suspended in a liquid, e.g., a slurry, from one location to another. Both internal and external cracking can result in structural failure of the industrial piping system and possibly decrease the service life of the equipment. The chaos and complexity associated with the uncertain behaviour inherent in pipeline systems lead to difficulty in detection and localisation of leaks in real time. The timely detection of leakage is important in order to reduce the loss rate and serious environmental consequences. The objective of this paper is to propose a new leak detection method based on an autoregressive with exogenous input (ARX) Laguerre fuzzy proportional-integral-derivative (PID) observation system. The objective of this paper is to propose a new leak detection method based on an autoregressive with exogenous input (ARX) Laguerre fuzzy proportional-integral-derivative (PID) observation system. In this work, the ARX–Laguerre model has been used to generate better performance in the presence of uncertainty. According to the results, the proposed technique can detect leaks accurately and effectively.

**Keywords:** autoregressive with exogenous input Laguerre (ARX–Laguerre); fuzzy; pipeline; PID; controller; PID observer

## 1. Introduction

Pipelines are the safest way for transporting crude oil, petroleum products, and natural gas over long distances. Pipelines deliver clear benefits in supporting economic growth as they provide a cheaper means to transport. However, oil and gas pipelines may be significantly damaged due to internal and external defects (e.g., corrosion, dents, gouges, weld defects). Construction and operational defects of pipes can pose major risks to supplies. Pipeline safety is possible using inspection and monitoring techniques which can be either internal or external in nature.

Over the last few years, a number of technologies have been reported to monitor pipelines such as acoustic emission [1–3], fibre optic sensor [4,5], digital signal processing, and mass–volume balance [6]. In [7], a real-time transient modelling method has been utilised for leakage detection and localization in the pipeline systems. In [8], an extended version of a real-time transient modelling method to estimate two leaks simultaneously in a piping system is proposed. The acoustic pulse reflectometry method has been used successfully to identify damage in pipelines utilising the time domain [9].

In [10], the cepstrum analysis technique is utilised to identify leaks in pipes. In [11], a new method based on auxiliary mass spatial probing by the stationary wavelet transform is suggested to detect damage in beams. Artificial intelligence with fuzzy logic has become the most effective approach, which attracts many investigators to deeply research it [12–16]. It has been successfully used for leak detection. In [17], a low-cost wireless sensor system

is introduced to detect of leaks in metallic piping systems. In [18,19], a neural network technique was utilized to detect the leak in a pipeline and has provided promising results. In [20], an artificial neural network was utilized to detect the leak in a pipeline such that the sound noise data were gathered through several microphones placed within a specific distance from the damaged part. The fast Fourier transform algorithm has been performed on data and supplied to a feed-forward network for making a final decision. In [11,21], a neural network technique was used for pattern recognition in oil pipe networks.

Various researchers have used observational approaches for fault diagnosis in pipes that are based on different algorithms [22–24]. The authors in [25] focused on leakage reconstruction in pipe systems utilising sliding mode observer. The authors in [26–28] focused on leakage reconstruction in pipe systems utilising a PID (proportional-integral-derivative) model and observer. In [29], a fuzzy PI observer was used to detect leaks in pipeline. In [30], a leak inspection device consisting of an adaptive Luenberger-type observer based upon a set of two-coupled partial differential equations governing the flow dynamics is proposed. To improve the input and output performance of ARX, in [26,31], the Laguerre method is applied to ARX to filter the input and output. In [27,29], a fuzzy PID observation method using the ARX–Laguerre technique is used for diagnosing fault in pipe.

The object of this paper is to develop a new technique based on autoregressive with exogenous input Laguerre (ARX–Laguerre) fuzzy PID to detect leaks in a pipe. For this aim, in the first step, the ARX–Laguerre technique is used for pipeline modelling. In the second step, the PID observer based on the ARX–Laguerre model is designed to detect leakage in the presence of uncertainties. The numerical results demonstrate that the proposed technique detects and estimates leaks accurately. The remainder part of this paper is organized as follows: in Section 2, the pipeline model equations are described using the momentum and continuity equations. The pipeline model equations based on the ARX–Laguerre technique are given in the Section 3. The proposed new technique based on ARX–Laguerre fuzzy PID observer to detect and locate leaks in a pipe is given in Section 4. The algorithm and simulation results analysis is given in Section 5. Moreover, in this section, the proposed method is compared with some other existing methods to illustrate its value. Finally, conclusions are given.

## 2. Pipeline Modelling

Here, we do not consider convective speed changes and compressibility effects in process lines. The mass flow rate ($m$), the length of the pipe ($\Gamma$), the flow in a pipe system ($\Phi$), the inlet pressure ($\wp_i$), and outlet pressure ($\wp_o$) at pipeline are assumed to be computable. Furthermore, the area of cross section ($a$) is fixed along the pipe. The suggested pipeline architecture is illustrated in Figure 1.

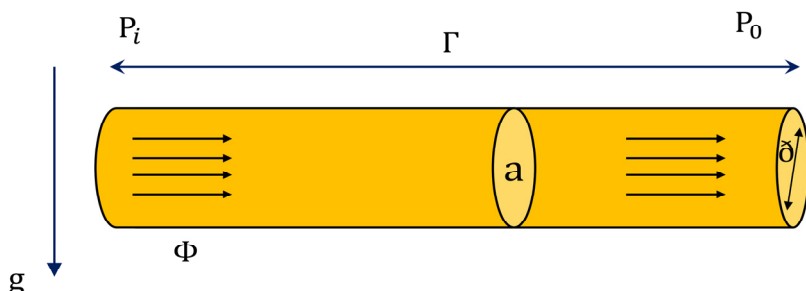

**Figure 1.** The suggested pipeline architecture.

The differential equation describing the dynamic behaviour of a fluid in a duct is based on the mass, momentum, and the conservation of energy [32]. Newton's second law of motion, when implemented to a control volume, generated the following momentum equation [32,33]:

$$\frac{\partial v}{\partial t} + \frac{1}{\rho}\frac{\partial \wp}{\partial x} + \frac{\Im}{2\eth}v^2 = 0 \tag{1}$$

If we substitute $v = \frac{\Phi}{a}$ as well as $\wp = \rho g \mathcal{H}$ in (1) the resulting momentum equation will be:

$$\frac{\partial \Phi}{a \partial t} + g \frac{\partial}{\partial x} \mathcal{H} + \frac{\Im \Phi^2}{2 \eth a^2} = 0 \tag{2}$$

Thus,

$$\frac{\partial \Phi}{\partial t} + ag \frac{\partial}{\partial x} \mathcal{H} + \frac{\Im \Phi^2}{2 \eth a} = 0 \tag{3}$$

in which $\mathcal{H}$ represents the pressure head, $\Phi$ the rate of flow in a pipe, $x$ the length of pipe, $t$ time steps, $g$ the gravity, $a$ the pipe cross-sectional area, $\eth$ the inside diameter of the pipe, and $\Im$ the pipe friction factor.

Coefficient of friction is typically assumed to be constant. In general, it was found to be a function of the Reynolds number ($Re$) and the pipe material roughness coefficient ($e$). The Swamee–Jain equation can be used to describe the friction factor for a pipe of circular section ($\eth$) as follows [34,35]:

$$\Im = \left( \frac{0.5}{\ln\left[0.27\left(\frac{e}{\eth}\right) + 5.74 \frac{1}{Re^{0.9}}\right]} \right)^2 \tag{4}$$

where $\Im$ is the pipe friction factor, $\eth$ is the inside diameter of the pipe and the pipe material roughness coefficient ($e$)

Reynolds number equation is determined via the following equation [36]:

$$Re = 4 \frac{\rho \Phi}{\pi \eth \mu} = \frac{\rho v \eth}{\mu} \tag{5}$$

in which $\rho$ represents the fluid density, and $\mu$ the viscosity of the flowing fluid. For $10^{-8} < \frac{e}{\eth} < 0.01$ as well as $5000 < Re < 10^8$ are provably correct.

$$\frac{\partial \wp}{\partial t} + \rho a^2 \frac{\partial v}{\partial x} = 0 \tag{6}$$

After applying the overall mass balance as well as the Reynolds transport theorem to the control volume the continuity equation will be obtained:

$$\frac{\partial \wp}{\partial t} + \rho a^2 \frac{\partial v}{\partial x} = 0 \tag{7}$$

The following equation can be acquired if we substitute the pressure head ($\mathcal{H}$) as well as the flow rate ($\Phi$) in Equation (7):

$$\frac{\partial \mathcal{H}}{\partial t} + \frac{a^2}{ga} \frac{\partial \Phi}{\partial x} = 0 \tag{8}$$

in which $a$ represents the speed of the wave inside a fluid filled elastic duct. The wave velocity depends on the elastic properties of the fluid and pipe. The pressure head ($\mathcal{H}$) and flow rate ($\Phi$) change as functions of position and time, $\mathcal{H}(x,t)$ and $\Phi(x,t)$, respectively, so that $x \in [0, \Gamma]$, where $\Gamma$ represents the length of the duct.

Now we can create a model of the pipe applying Equations (3) and (8). These equations need to be solved; however, coming to analytical solutions is not easy. Because of this, different methods need to be used to solve these equations such as characteristics and finite difference approaches [37]. Here, the finite difference approach is implemented such that Equations (3) and (8) are discretized to obtain a system of ordinary differential equations. The considered finite difference approach discretizes the whole pipe into $N$ smaller sections [37,38]. Finite difference technique with a fixed step size $\Delta s$ is, historically, the most popular time-stepping approach. Here, we consider finite difference method

because it is an easy-to-use approach and is specially designed and applied for nonlinear observer models. In this study we define it as follows:

$$\frac{\partial \Phi(s_{i-1},t)}{\partial s} \approx \frac{\Delta \Phi(s_{i-1},t)}{\Delta s} \approx \frac{\Phi_i - \Phi_{i-1}}{\Delta s}$$
$$\frac{\partial \mathcal{H}(s_{i-1},t)}{\partial s} \approx \frac{\Delta \mathcal{H}(s_{i-1},t)}{\Delta s} \approx \frac{\mathcal{H}_i - \mathcal{H}_{i-1}}{\Delta s} \tag{9}$$

$\forall i = 1, \cdots, n$, in which $n$ represents the number of points of the grid, and $\Delta s = s_{i+1} - s_i$ represents the size of the $i$-section between the two successive grid points. The computational domain $s \in [0, \Gamma]$ is divided up into three smaller domains, $\{s_k\} := \{0, s_{leak}, \Gamma\}$, so that $s_{leak}$ indicates the location of leak; see Figure 2. The leak flow rate can be measured by $\Phi_{leak} = C_d a_{leak} \sqrt{2g} \sqrt{\mathcal{H}(s_{leak}, t)}$, such that $C_d$ represents efflux coefficient, and $a_{leak}$ the cross-sectional area along the leak path. The leak flow rate can be calculated by $\Phi_{leak} = \Lambda \sqrt{\mathcal{H}(s_{leak}, t)}$, in which $\Lambda = C_d a_{leak} \sqrt{2g}$. The behaviour of a dynamic pipeline network can be described by an ordinary differential equation system:

$$\dot{\Phi}_1 = \frac{ga}{s}(\mathcal{H}_1 - \mathcal{H}_2) - \frac{\Im \Phi_1^2}{2\partial a}$$
$$\dot{\mathcal{H}}_{leak} = \frac{c^2}{gas}(\Phi_1 - \Phi_2 - \Lambda\sqrt{\mathcal{H}_{leak}}) \tag{10}$$
$$\dot{\Phi}_2 = \frac{ga}{\Gamma - s}(\mathcal{H}_2 - \mathcal{H}_3) - \frac{\Im \Phi_2^2}{2\partial a}$$

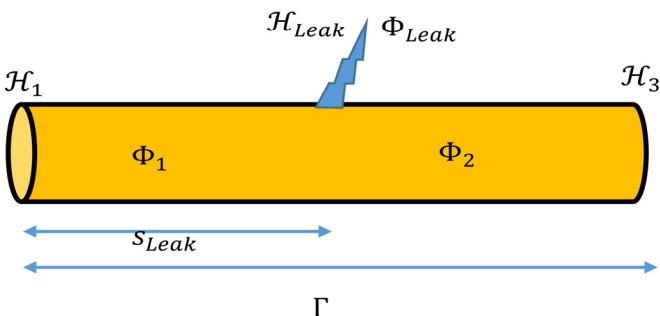

**Figure 2.** The suggested pipeline architecture.

Suppose that both inlet and outlet pressures, $\mathcal{H}_1$ and $\mathcal{H}_3$, respectively, are known and have been defined using external means such as a pump. The pressure $\mathcal{H}_2$ and the inlet and outlet flow rate ($\Phi_1$ and $\Phi_2$, respectively) of the leakage point are considered to be variables. From the continuity equation we can write:

$$\Phi_1 = \Phi_{leak} + \Phi_2 \tag{11}$$

## 3. Pipeline Modelling Based on the ARX–Laguerre Technique

For many years, pipelines played a huge role in oil and gas industries, as they significantly reduce transport costs. Leakage inspection in transmission pipelines is crucially significant for safe operation. In general, there are various fault detection methods, each with different potentials; however, the selection of proper leak detection technique is difficult. This is particularly important when they deal with various types of uncertain conditions. To deal with this problem, we introduce a fuzzy ARX–Laguerre PID observer in Section 4. First, in this study, the ARX–Laguerre technique is used for pipeline modelling. In the second step, the PID observer based on the ARX–Laguerre model is designed to detect leakage in the presence of uncertainties. The proposed model-based ARX–Laguerre orthonormal method is represented by developing its coefficients associated to the flow input and flow output, Fourier coefficients, and Laguerre-based orthonormal function, as follows [23,39]:

$$M_0(s) = \sum_0^{i_a} \lambda_{n.a}\left(\sum_{j=1}^{\infty} \ell_a * M_0(s)\right).x_{n.M_0}(s) + \sum_0^{i_b} \lambda_{n.b}\left(\sum_{j=1}^{\infty} \ell_b * M_i(s)\right).x_{n.M_i}(s) \tag{12}$$

in which $M_0(s)$, ($\lambda_{n.a}$ and $\lambda_{n.b}$), ($i_a, i_b$) ($\ell_a, \ell_b$),*, $M_i(s)$, $x_{n.M_0}(s)$, and $x_{n.M_i}(s)$ represent the pipe outflow, Fourier coefficients, the order of the system, Laguerre orthonormal function, convolution product, pipe inflow, exhaust filter, and entrance filter, respectively. By expanding the ARX model on Laguerre orthonormal bases, the following state-space model can be obtained:

$$\begin{cases} M(s+1) = \left[AM(s) + B_y(y(s) + \alpha_s(k)) + B_u(u(s) + a_p(s))\right] \\ \quad y(s) = (S)^T M(s) + B_s \alpha_s(s), \end{cases} \tag{13}$$

in which $M(s), y(s), u(s), \alpha_p(s)$, and $\alpha_s(s)$ represent the state vector, calculated output, control input, pump defect, and sensor defect, respectively. $A, B_y, B_u$, and $B_s$, as well as $S$, represent matrices of coefficients.

## 4. ARX–Laguerre Fuzzy PID Observation Technique

In this section, the ARX–Laguerre fuzzy PID observation system is proposed to detect and estimate a leak in pipelines.

### 4.1. Modelling of Dynamic System by ARX–Laguerre

Let us consider the linear ARX state space model with disturbances illustrated by the following equation to formulate the dynamic fault detection problem:

$$\begin{cases} M(s+1) = \left[A\,M(s) + B_u(u(s) + \alpha_p(s))\right] \\ \quad y(s) = (S)^T M(s) + B_s \alpha_s(s), \end{cases} \tag{14}$$

We define the ARX model on Laguerre base as follows [31,40]:

$$\begin{aligned} y(k) &= \sum_0^{N_a-1} S_{(n,p)} x_{(n,y)}(s) + \sum_0^{N_b-1} S_{(n,b)} x_{(n,u)}(s) \\ X(k) &= \left[\begin{array}{cc} x_{(n,u)}(s) & x_{(n,y)}(s) \end{array}\right] \\ x_{(n,y)}(s) &= L_n^a(k, \xi_p) * y(s) \\ x_{(n,u)}(s) &= L_n^b(k, \xi_b) * u(s) \end{aligned} \tag{15}$$

in which $y(k)$, $u(k)$, $\left(K_{(n,p)}, K_{(n,s)}\right)$, $(N_a, N_b), x_{(n,y)}(s), x_{(n,u)}(s)$, and $\left(L_n^a(s, \xi_a), L_n^b(s, \xi_b)\right)$ represent the pipe outflow, pipe inflow, Fourier coefficients, exhaust filter, entrance filter, and Laguerre orthonormal function, respectively.

Using Equation (16) the following state-space model can be obtained in the presence of failures of the pump and sensor as well as disturbances:

$$\begin{cases} M_p(s+1) = \left[AM_p(s) + B_y(y_p(s) + \alpha_s(s)) + B_u(u(s) + \alpha_p(s))\right] \\ \quad y(s) = (S)^T M_p(s) + B_s \alpha_s(s), \end{cases} \tag{16}$$

The fault of the pump is calculated using the following formula:

$$e_M(s) = \begin{array}{c} e_y(s) = y_p(s) - y(s) \\ \left[\begin{array}{c} x_{p(n,u+\alpha_p)}(s) - x_{(n,u)}(s) \\ x_{p(n,y_a)}(s) - x_{(n,y)}(s) \end{array}\right] \end{array} \tag{17}$$

such that

$$x_{a(n,u+\alpha_p)}(s) \neq M_{(n,u)}(s) \to M_p(s) \neq M(s) \to y_p(s) \neq y(s) \to e_y(s) \neq 0 \tag{18}$$

The fault of the sensor is calculated using the following formula:

$$e_M(s) = \begin{bmatrix} e_y(s) = y_p(s) - y(s) \\ M_{p(n,u)}(k) - M_{(n,u)}(s) \\ M_{p(n,y_p+\alpha_s)}(k) - M_{(n,y)}(s) \end{bmatrix} \tag{19}$$

such that

$$M_{p(n,y_p+\alpha_s)}(s) \neq x_{(n,y)}(s) \rightarrow M_p(s) \neq M(s) \rightarrow y_p(s) \neq y(s) \rightarrow e_y(S) \neq 0. \tag{20}$$

*4.2. Fault Diagnosis*

In this study, the ARX–Laguerre fuzzy PID observation system is proposed to identify pump and sensor defects in pipes. We define the proposed technique by the following formulas in the presence of failures of the pump and sensor in the pipe:

$$\begin{cases} \hat{M}(s+1) = A\hat{M}(s) + B_y(\hat{y}(s) + \hat{\alpha}_s(s)) + B_y(u(s) + \hat{\alpha}_p(s)) + K_p e(s) \\ e_s(s) = (q_s(s) - \hat{q}_s(s)) \\ e_p(s) = (w_p(s) - \hat{w}_p(s)) \\ \hat{\alpha}_p(s+1) = \hat{\alpha}_p(s) + K_{ip}e_p(s) + K_{d_p}\big(e_p(s+1) + e_p(s) + e_p(s-1)\big) \\ \hat{\alpha}_s(s+1) = \hat{\alpha}_s(s) + K_{is}e_s(s) + K_{d_s}(e_s(s+1) + e_s(s) + e_s(s-1)) \\ \hat{y}(s+1) = (S)^T \hat{M}(s+1) + \beta_s \hat{\alpha}_s(s) \end{cases} \tag{21}$$

where $\hat{M}(s)$ represents the state vector, $\alpha_p(s)$ pump defect, $\alpha_s(s)$ sensor defect, and $\hat{y}(s)$ the output of the system, and $A$, $B_y$, $B_u$, and $B_s$, as well as $S$, represent matrices of coefficients. In accordance with Equation (21), in this paper, we particularly study three main cases and types of faults in pipe.

Case 1: In case $\alpha_p \neq 0$, $\alpha_s = 0$, and $\hat{\alpha}_p(s) \neq \alpha_p(s)$, we have:

$$(y(s+1) - \hat{y}(s+1) \neq 0) \,\&\, \big(M(s+1) - \hat{M}(s+1)\big) \neq 0 \implies$$
$$\begin{bmatrix} M_1^T(s+1) & M_2^T(s+1) \end{bmatrix}^T - \begin{bmatrix} \hat{M}_1^T(s+1) & \hat{M}_{2,\alpha_p}^T(s+1) \end{bmatrix}^T \neq 0 \implies$$
$$\begin{cases} M_{(n,u)}(s) - \hat{M}_{(n,u+\alpha_p)}(s) \neq 0 \\ M_{(n,y)}(s) - \hat{M}_{(n,y)}(s) \neq 0 \end{cases} \tag{22}$$

In case $\alpha_p \neq 0$, $\alpha_s = 0$, and $\hat{\alpha}_p(s) = \alpha_p(s)$, we have:

$$(y(s+1) - \hat{y}(s+1) = 0) \,\&\, \big(M(s+1) - \hat{M}(s+1)\big) \neq 0 \implies$$
$$\begin{bmatrix} M_1^T(s+1) & M_2^T(s+1) \end{bmatrix}^T - \begin{bmatrix} \hat{M}_1^T(s+1) & \hat{M}_{2,\alpha_p}^T(s+1) \end{bmatrix}^T \neq 0 \implies$$
$$\begin{cases} M_{(n,u)}(s) - \hat{M}_{(n,u+\alpha_p)}(s) \neq 0 \\ M_{(n,y)}(s) - \hat{M}_{(n,y)}(s) = 0 \end{cases} \tag{23}$$

In accordance with Equations (22) and (23), in case the error related to ARX–Laguerre fuzzy PID technique is close to zero, the detection rate of defect is very high.

The following formula can be defined for fault in the pump:

$$\hat{\alpha}_p = \alpha_q \rightarrow q_p - \hat{q}_p \cong 0 \,\&\, w - \hat{w} \neq 0 \rightarrow r = w - \hat{w} \tag{24}$$

Case 2: In case $\alpha_s \neq 0$, $\alpha_p = 0$, and $\hat{\alpha}_s(s) \neq \alpha_s(s)$, we have:

$$(y(s+1) - \hat{y}(s+1) \neq 0) \,\&\, \big(M(s+1) - \hat{M}(s+1)\big) \neq 0 \implies$$
$$\begin{bmatrix} M_1^T(s+1) & M_2^T(s+1) \end{bmatrix}^T - \begin{bmatrix} \hat{M}_1^T(s+1) & \hat{M}_{2,\alpha_p}^T(s+1) \end{bmatrix}^T \neq 0 \implies$$
$$\begin{cases} M_{(n,u)}(s) - \hat{M}_{(n,u)}(s) \neq 0 \\ M_{(n,y)}(s) - \hat{M}_{(n,y+\alpha_s)}(s) \neq 0 \end{cases} \tag{25}$$

In case $\alpha_s \neq 0$, $\alpha_p = 0$, and $\hat{\alpha}_s(s) = \alpha_s(s)$, we have:

$$
\begin{gathered}
(y(s+1) - \hat{y}(s+1) = 0) \ \& \ \left(M(s+1) - \hat{M}(s+1)\right) \neq 0 \implies \\
\left[ \ M_1^T(s+1) \quad M_2^T(s+1) \ \right]^T - \left[ \ \hat{M}_1^T(s+1) \quad \hat{M}_{2,\alpha_p}^T(s+1) \ \right]^T \neq 0 \implies \\
\begin{cases}
M_{(n,u)}(s) - \hat{M}_{(n,u)}(s) = 0 \\
x_{(n,y)}(s) - \hat{x}_{(n,y+\alpha_s)}(s) \neq 0
\end{cases}
\end{gathered}
\tag{26}
$$

In accordance with Equations (25) and (26), the ARX–Laguerre fuzzy PID has a significant influence on the efficiency of sensor defect detection in a duct.

The following formula can be defined for fault in sensor:

$$
\hat{\alpha}_s = \alpha_s \rightarrow w - \hat{w} \cong 0 \ \& \ q_p - \hat{q}_p \neq 0 \rightarrow r = q_p - \hat{q}_p
\tag{27}
$$

Case 3: In case $\alpha_s \neq 0$, $\alpha_p = 0$, $\hat{\alpha}_s(s) \neq \alpha_s(s)$, and $\hat{\alpha}_p(s) \neq \alpha_p(s)$ we have:

$$
\begin{gathered}
(y(s+1) - \hat{y}(s+1) \neq 0) \ \& \ \left(M(s+1) - \hat{M}(s+1)\right) \neq 0 \implies \\
\left[ \ M_1^T(s+1) \quad M_2^T(s+1) \ \right]^T - \left[ \ \hat{M}_{1,\alpha_s}^T(s+1) \quad \hat{M}_2^T(s+1) \ \right]^T \neq 0 \implies \\
\begin{cases}
M_{(n,u)}(s) - \hat{M}_{(n,u+\alpha_p)}(s) \neq 0 \\
M_{(n,y)}(s) - \hat{M}_{(n,y+\alpha_s)}(s) \neq 0
\end{cases}
\end{gathered}
\tag{28}
$$

In case $\alpha_s \neq 0$, $\alpha_p = 0$, and $\hat{\alpha}_s(s) = \alpha_s(s)$, we have:

$$
\begin{gathered}
(y(s+1) - \hat{y}(s+1) \neq 0) \ \& \ \left(M(s+1) - \hat{M}(s+1)\right) \neq 0 \implies \\
\left[ \ M_1^T(s+1) \quad M_2^T(s+1) \ \right]^T - \left[ \ \hat{M}_1^T(s+1) \quad \hat{M}_{2,\alpha_p}^T(s+1) \ \right]^T \neq 0 \implies \\
\begin{cases}
M_{(n,u)}(s) - \hat{M}_{(n,u+\alpha_p)}(s) \neq 0 \\
M_{(n,y)}(s) - \hat{M}_{(n,y+\alpha_s)}(s) \neq 0
\end{cases}
\end{gathered}
\tag{29}
$$

In accordance with Equation (29), in case the pipe includes sensor and pump failures, the signals received from pump and joint variable can identify the defects. Signal sensor and pump faults are:

$$
\hat{\alpha}_p = \alpha_p \ \& \ \hat{\alpha}_s = \alpha_s \rightarrow r_1 = w - \hat{w} \gg 0 \ \& \ r_2 = q_p - \hat{q}_p \gg 0
\tag{30}
$$

To increase the signal estimation accuracy and to modify the performance of fault estimation of the ARX–Laguerre PID technique, optimal fuzzy observer coefficients, $K_{p_p}$, $K_{i_p}$, $K_{d_p}$, $K_{p_s}$, $K_{i_s}$ and $K_{d_s}$, are applied, which are defined as follows:

$$
\begin{aligned}
K_{i_p} &= \frac{K_{p_p}}{T_{i_p}}, \ K_{d_p} = K_{p_p}.T_{d_p} \\
K_{i_s} &= \frac{K_{p_s}}{T_{i_s}}, \ K_{d_s} = K_{p_s}.T_{d_s}
\end{aligned}
\tag{31}
$$

where $T_{i_p}$, $T_{i_s}$, $T_{d_p}$, and $T_{d_s}$, represent the integral gain for pump failure, the integral gain for sensor failure, the derivative gain for pump failure, and the derivative gain for sensor failure, respectively. Following Equation (29), we have:

$$
\begin{aligned}
\beta_p &= \frac{T_{i_p}}{T_{i_p}}, \ K_{ip} = \frac{\left(K_{p_p}\right)^2}{\beta_p K_{d_p}} \\
\beta_s &= \frac{T_{i_s}}{T_{i_s}}, \ K_{i_s} = \frac{\left(K_{p_s}\right)^2}{\beta_s K_{d_s}}
\end{aligned}
\tag{32}
$$

Normalization of the above equation can be performed by the formula described below:

$$K'_{pp} = \frac{K_{pp} - K_{pp(min)}}{K_{pp(max)} - K_{pp(min)}} \in [0,1], \ K'_{pp} = \frac{K_{dp} - K_{dp(min)}}{K_{pp(max)} - K_{pp(min)}} \in [0,1], 2 \le \beta_p \le 5$$
$$K'_{ps} = \frac{K_{ps} - K_{ps(min)}}{K_{ps(max)} - K_{ps(min)}} \in [0,1], \ K'_{ds} = \frac{K_{ds} - K_{ds(min)}}{K_{ds(max)} - K_{ds(min)}} \in [0,1], 2 \le \beta_s \le 5 \tag{33}$$

such that $\beta = \frac{\sum_i \alpha(x_i) \cdot x_i}{\sum_i \alpha(x_i)}$ represents a membership function.

## 5. Simulation Results

In this section, we evaluate our proposed technique on a pipe model under the leak condition in the presence of failures of the pump and sensor in the pipe. In order to check the efficiency of the proposed ARX–Laguerre fuzzy PID observation technique for fault detection in the pipe, we consider two cases, pipe with fault and under no-fault conditions.

Pipe under no-fault condition. In this case, the duct functions under optimal circumstances and performs well. The input–output signals of the pipe model in a healthy state can be computed as follows:

$$r(w) = w - \hat{w} \rightarrow r(w) = w - \left(w_{Observer} + \alpha_p\right) \rightarrow r(w) = w - \left(w_{Observer} + 0\right) \cong 0$$
$$r(\phi) = \phi - \hat{\phi} \rightarrow r(\phi) = \phi - \left(\phi_{Observer} + \alpha_s\right) \rightarrow r(\phi) = \phi - \left(\phi_{Observer} + 0\right) \cong 0 \tag{34}$$

The sensor signal for the pipe under no-fault condition and no noise impact is shown in Figure 3. The pump signal for the pipe under no-fault condition and no noise impact is shown in Figure 4.

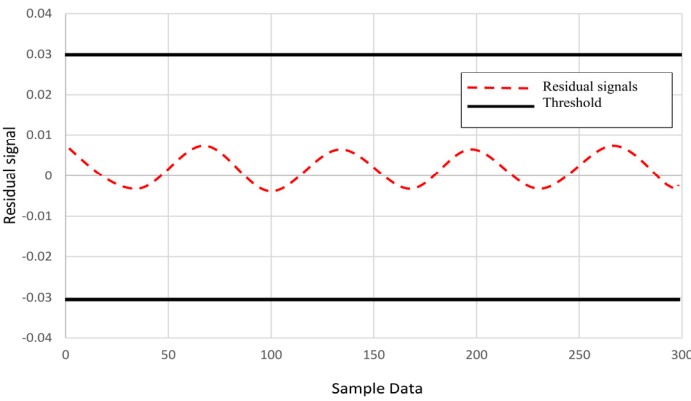

**Figure 3.** The sensor signal for the pipe under no-fault condition.

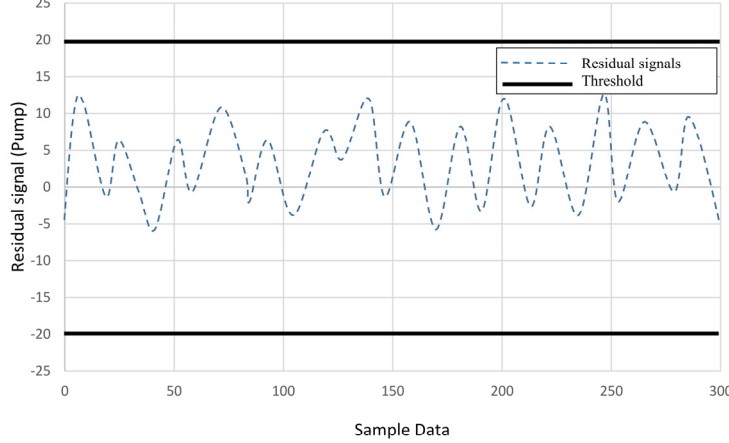

**Figure 4.** The pump signal for the pipe under no-fault condition.

For a healthy system, the pump and sensor faults can be described as follows:

$$Fault_{pump} = \alpha_p(t - t_0) = \begin{cases} 0, & t < t_p \\ \alpha_p, & t > t_p \end{cases}$$

$$Fault_{sensor} = \alpha_s(t - t_0) = \begin{cases} 0, & t < t_s \\ \alpha_s, & t > t_s \end{cases} \quad (35)$$

The effectiveness of the proposed technique for fault estimation under no-fault condition is shown Figure 5. As can be observed from Figure 5, the proposed method is more effective than the ARX–Laguerre PI observer [41] and the adaptive fuzzy observer [42]. The error between the predicted output and the expected output based on the proposed technique under no-fault condition is shown in Figure 6. It can be seen that the proposed method yields more accurate results compared with ARX–Laguerre PI observer [41] and the adaptive fuzzy observer [42].

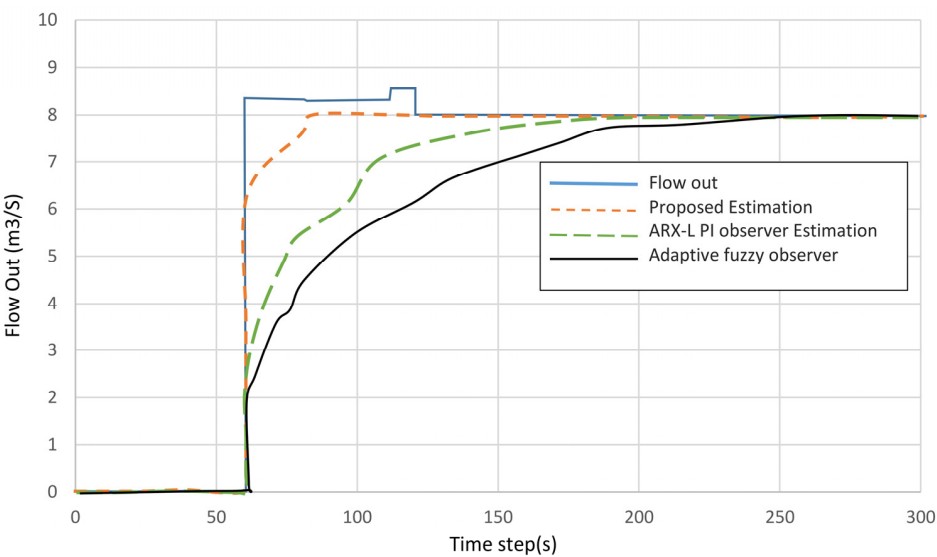

**Figure 5.** The effectiveness of the proposed technique for fault estimation under no-fault condition.

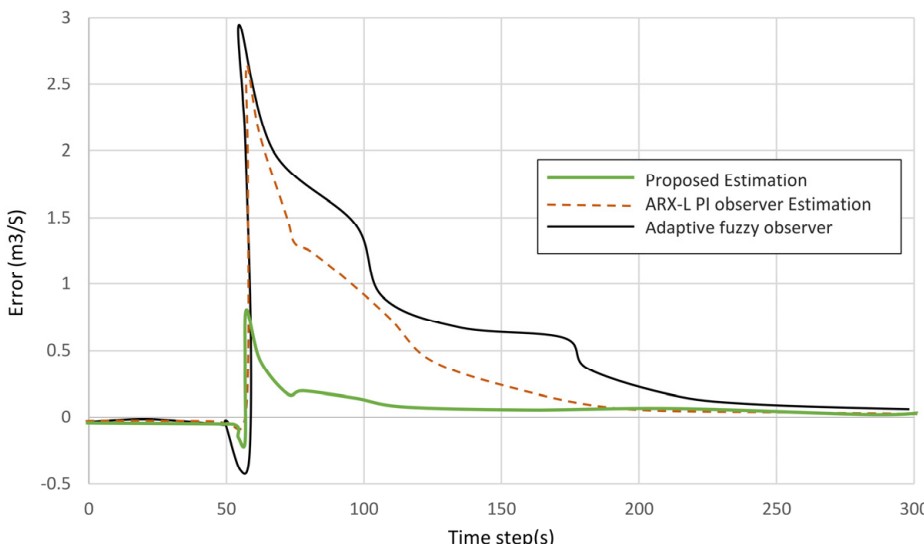

**Figure 6.** The error between the predicted output and the expected output based on the proposed technique under no-fault condition.

Pipe under fault condition. In this case, the duct functions under faulty circumstances. The duct has two kinds of defects simultaneously, the sensor defect and the pump defect.

The input–output signals from sensor and pump in the pipe with a fault state can be computed as follows:

$$r(w) = w - \hat{w} \rightarrow r(w) = w - (w_{Observer} + \alpha_p) \gg 0$$
$$r(\phi) = \phi - \hat{\phi} \rightarrow r(\phi) = \phi - (\phi_{Observer} + \alpha_s) \gg 0$$

(36)

where

$$w_{1\alpha_p}(m) = \begin{cases} 55, & 10 \leq t \leq 25 \\ 0, & otherwise \end{cases}$$
$$\phi_{1\alpha_s}(m) = \begin{cases} 0.6, & 10 \leq t \leq 25 \\ 0, & otherwise \end{cases}$$

(37)

The pump signal for the pipe under fault condition and no noise impact is shown in Figure 7. The sensor signal for the pipe under fault condition is shown in Figure 8.

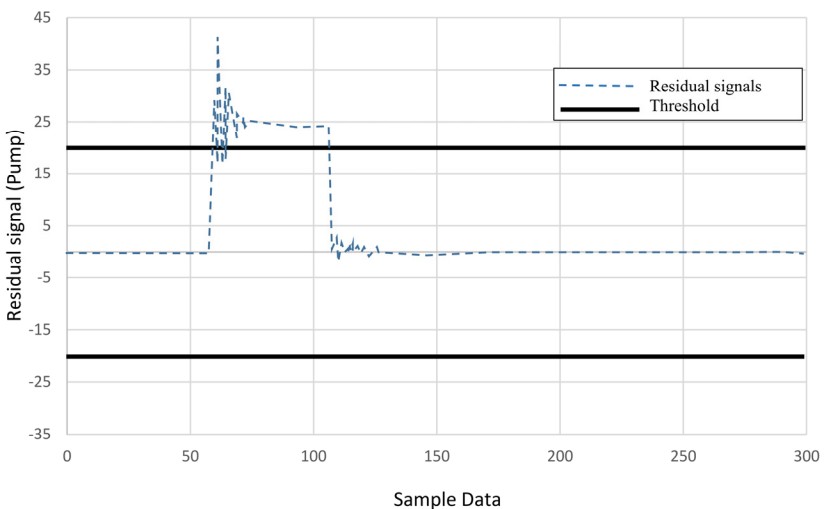

**Figure 7.** The pump signal for the pipe under fault condition.

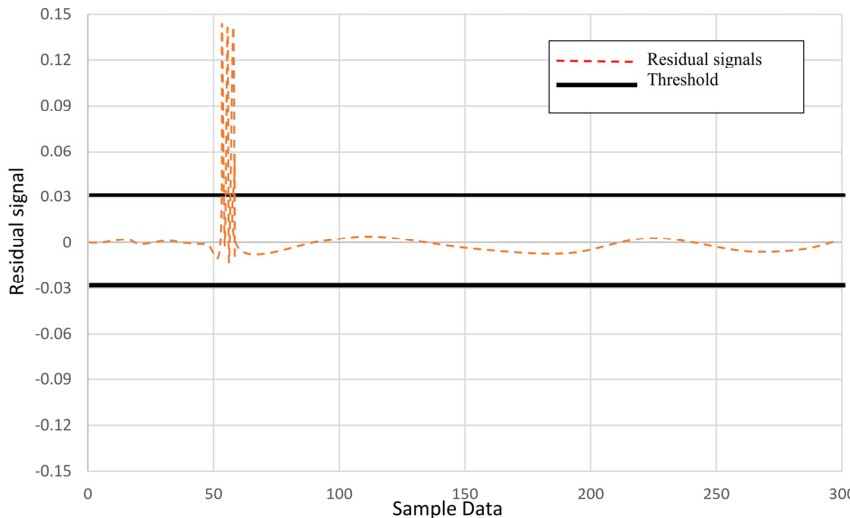

**Figure 8.** The sensor signal for the pipe under fault condition.

The effectiveness of the proposed technique for fault estimation under fault condition is shown Figure 9. As can be observed from Figure 9, the proposed method is more effective than the ARX–Laguerre PI observer [41] and the adaptive fuzzy observer [42]. The error between the predicted output and the expected output based on the proposed technique under fault condition is shown in Figure 10. It can be seen that the proposed method yields

more accurate results compared with ARX–Laguerre PI observer [41] and the adaptive fuzzy observer [42]. Furthermore, the delay for the proposed method to fault detection in both Figures 9 and 10 is less than the other methods.

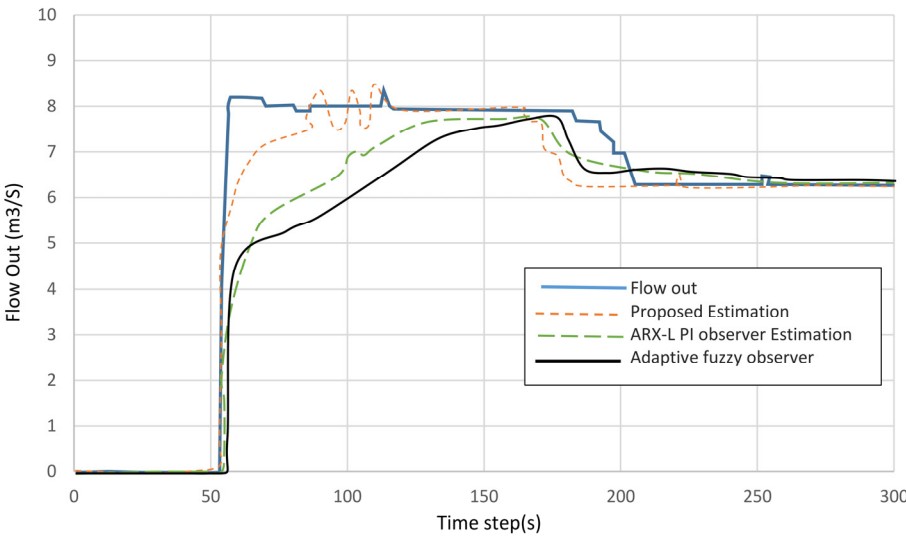

**Figure 9.** The effectiveness of the proposed technique for fault estimation under fault condition.

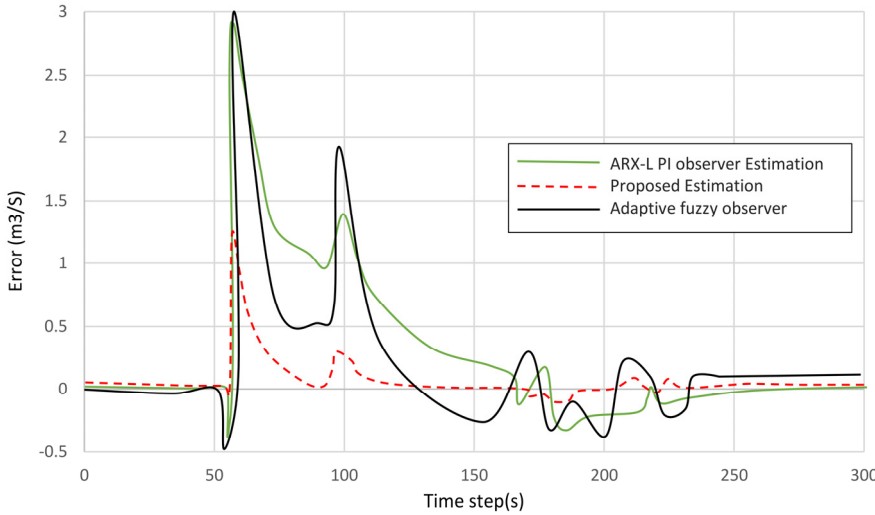

**Figure 10.** The error between the predicted output and the expected output based on the proposed technique under fault condition.

The effectiveness of the proposed technique for fault estimation at leakage point is shown in Figure 11. It can be seen from this figure that our proposed method detects fault in less time in comparison with ARX–Laguerre PI observer [41] and the adaptive fuzzy observer [42].

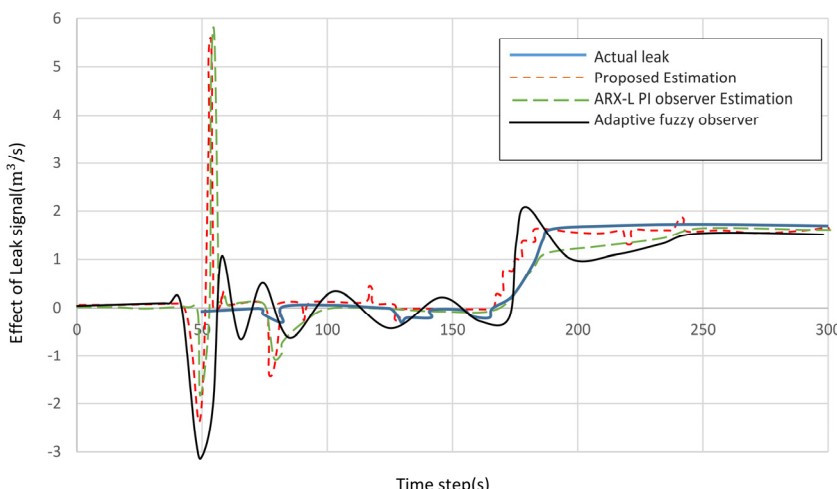

**Figure 11.** The effectiveness of the proposed technique for fault estimation at leakage point in pipe.

## 6. Conclusions

The task of precise defect detection in the pipeline system is a formidable challenge due to the uncertainties in leak signal. To better deal with uncertainties in the leak signal, in this paper, an ARX–Laguerre PID-observer is introduced to perform fault diagnosis in the pipeline system. First, in this study, the ARX–Laguerre technique was used for pipeline modelling. In the second step, the PID observer based on the ARX–Laguerre model was designed to detect leakage in the presence of uncertainties. The performance of the proposed algorithm was tested in numerical simulations. According to the results, the proposed technique can accurately locate the leakage point. Despite the high accuracy of the proposed fault diagnosis method, it has a disadvantage of large extensive computation. In the future, the proposed observation method will be used to enhance the performance of fault diagnosis when the uncertainties are in the form of Z-numbers.

**Author Contributions:** Conceptualization, R.J., S.R., C.V.-J., A.G. and F.A.; methodology, R.J., S.R., C.V.-J., A.G. and F.A.; software, S.R.; validation S.R.; formal analysis, R.J. and S.R.; investigation, R.J. and S.R.; resources, R.J., S.R., C.V.-J., A.G. and F.A.; writing—original draft preparation, R.J., S.R., C.V.-J., A.G. and F.A.; writing—review and editing, R.J., S.R., C.V.-J., A.G. and F.A. All authors have read and agreed to the published version of the manuscript.

**Funding:** This research received no external funding.

**Institutional Review Board Statement:** Not applicable.

**Informed Consent Statement:** Not applicable.

**Data Availability Statement:** Not applicable.

**Conflicts of Interest:** The authors declare no conflict of interest.

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
