# Peer review of "Pipeline Leak Detection and Estimation Using Fuzzy PID Observer"

_electronics, doi:10.3390/electronics11010152_

Round 1

Reviewer 1 Report

In this manuscript (ms), the authors present an auto regressive with exogenous input (ARX)-Laguerre fuzzy proportional-integral-derivative (PID) observation system in order to detect and estimate a leak in pipelines. The results are remarkably interesting and potentially important.

The authors have successfully addressed most of my previous comments, but there are still some issues related with the presentation that could improve the ms. These are related with:

  • Equation (9) is the same as Equation (3) and the discussion in lines 114 to 117 rather complicate than help the reader.
  • The connection of the mathematical formulation of Sections 3 and 4, with that of Section 2. Such a connection in my opinion should be made before presenting the results in Section 5.
  • In the Section 5. Results comparison is made with the “ARX-Laguerre PID observer[26]” where Ref.[26] is entitled: “fault diagnosis of a robot manipulator based on an ARX-laguerre fuzzy PID observer”. The authors should definitely discuss, for the benefit of the reader, in Section 5 as well as in Section 6 the difference between their method and that of Ref.[26]. Otherwise, confusion arises.

The authors should treat these three important issues before I could suggest publication of the ms.

The following minor points need improvement:

  1. Define PID line 60.
  2. The length Γ is not defined correctly in line 80.
  3. On page 3, Equation (1) there appear two types of v. Please use just one.
  4. On page 3, Equation (6) \hat{a} appears which is not defined in the ms.
  5. In line 113, the quantity a is already defined as the cross-sectional area in line 82. Do mean \hat{a}?
  6. In line 233, “Figures 6” -> “in Figure 6”
  7. In line 236, “Figures 7” -> “Figure 7”
  8. In line 262, “Figures 10” -> “in Figure 10”
  9. In line 265, “Figures 11” -> “Figure 11”

Author Response

Dear Reviewer,

Thank you for your valuable comments on our paper.

Please, find attached our responses to these comments.

Kind regards,

Alexander Gegov

Reviewer 2 Report

Dear Authors,

Based on the first round review of the manuscript entitled: Pipeline leak detection and estimation using fuzzy PID observer, the reviewer has the following comments:

  1. Please more explain about the contributions and explain deeply the advantages of your manuscript compared with the following three research articles: 1) Piltan, Farzin, and Jong-Myon Kim. "Advanced fuzzy-based leak detection and size estimation for pipelines." Journal of Intelligent & Fuzzy Systems 38, no. 1 (2020): 947-961. , 2) Piltan, Farzin, and Jong-Myon Kim. "Pipeline Leak Detection and Estimation Using Fuzzy-Based PI Observer." In International Conference on Intelligent and Fuzzy Systems, pp. 1122-1129. Springer, Cham, 2019. and 3) Piltan, Farzin, Muhammad Sohaib, and Jong-Myon Kim. "Fault diagnosis of a robot manipulator based on an ARX-Laguerre fuzzy PID observer." In International Conference on Robot Intelligence Technology and Applications, pp. 393-407. Springer, Cham, 2017.
  2. In Figure 3, how it can generate the residual signal??? please explain it deeply.
  3. How you can find Figure 4,5, and 8? these Figures are very similar to Figures in the following reference: Piltan, Farzin, Muhammad Sohaib, and Jong-Myon Kim. "Fault diagnosis of a robot manipulator based on an ARX-Laguerre fuzzy PID observer." In International Conference on Robot Intelligence Technology and Applications, pp. 393-407. Springer, Cham, 2017.
  4. How you can find Figures 6 and 10? this figure is similar to the figure in the following reference: Piltan, Farzin, and Jong-Myon Kim. "Advanced fuzzy-based leak detection and size estimation for pipelines." Journal of Intelligent & Fuzzy Systems 38, no. 1 (2020): 947-961.
  5. What is the limitation of your technique? please add it in the conclusion.
  6. References didn't add into the manuscript. please solve this issue in the revised manuscript.

Regards,

Author Response

(The authors gave the same response as above.)

Reviewer 3 Report

No further comments.

Author Response

(The authors gave the same response as above.)

Reviewer 4 Report

Broad comments. Leakage detection in pipes is an issue of great importance, especially for the oil and gas industry. The authors have presented a concise overview of the topic and managed to illuminate the main advantages of the proposed system. As a general drawback, I would highlight the lack of references in relevant applied studies (e.g. 1), where in-situ solutions targeting similar applications have been introduced.  

[1] Christos, Spandonidis C., et al. "Autonomous low-cost Wireless Sensor platform for Leakage Detection in Oil and Gas Pipes." 2021 10th International Conference on Modern Circuits and Systems Technologies (MOCAST). IEEE, 2021.

Specific comments. In general, the text is well structured and has clearly defined topics. Some comments for improvement:

  1. The abstract should be refined such that it better describes the work performed. Authors could consider refining the structure of the abstract that initially presents the problem and afterward the proposed solution and the work performed, among the main outcomes.
  2. Authors should pay more effort into highlighting the motivation and mainly the innovation o their work. The last paragraph of section 1 should be descriptive of the objectives of the work.
  3. Authors are encouraged to justify the selection of their methodology.
  4. The signals presented in Figures 4, 5, and 8 are far from the actual signals one could see in the field. While assumptions are a crucial part of the research procedure authors could add noise or comment to their manuscript.
  5. In the same direction, while a demonstration of methods effectiveness is provided, there is one missing answer, that normally we ask ourselves in similar research activities: what is the added value of the method compared to using a simple threshold value for detection of leakage? Authors are encouraged to add a relevant discussion.
  6. Results regarding the computational effort could be fruitful.
  7. Since figures 10, 11, and especially 12 are the most important when it comes to the illustration of methods capacity, authors are encouraged to provide further explanation on what is shown.

Author Response

(The authors gave the same response as above.)

Round 2

Reviewer 2 Report

Dear Authors,

Thank you for your response letter. Regarding the second round review of the manuscript, the reviewer has the following comments

  1. Please more explain about the contributions and explain deeply the advantages of your manuscript compared with the following three research articles (not only add these papers as a reference): 1) Piltan, Farzin, and Jong-Myon Kim. "Advanced fuzzy-based leak detection and size estimation for pipelines." Journal of Intelligent & Fuzzy Systems 38, no. 1 (2020): 947-961. , 2) Piltan, Farzin, and Jong-Myon Kim. "Pipeline Leak Detection and Estimation Using Fuzzy-Based PI Observer." In International Conference on Intelligent and Fuzzy Systems, pp. 1122-1129. Springer, Cham, 2019. and 3) Piltan, Farzin, Muhammad Sohaib, and Jong-Myon Kim. "Fault diagnosis of a robot manipulator based on an ARX-Laguerre fuzzy PID observer." In International Conference on Robot Intelligence Technology and Applications, pp. 393-407. Springer, Cham, 2017.
  2. In Figure 3, how it can generate the residual signal??? please explain it deeply. (please show it in the figure as well, I think this figure is not correct)
  3. How you can find Figure 4,5, and 8? (I know that you found it using MATLAB but explain in more detail about the inputs and how you found it. ( these Figures are very similar to Figures in the following reference: Piltan, Farzin, Muhammad Sohaib, and Jong-Myon Kim. "Fault diagnosis of a robot manipulator based on an ARX-Laguerre fuzzy PID observer." In International Conference on Robot Intelligence Technology and Applications, pp. 393-407. Springer, Cham, 2017.)
  4. How you can find Figures 6 and 10? (such as above question you should explain the inputs and how you found it ) this figure is similar to the figure in the following reference: Piltan, Farzin, and Jong-Myon Kim. "Advanced fuzzy-based leak detection and size estimation for pipelines." Journal of Intelligent & Fuzzy Systems 38, no. 1 (2020): 947-961.

Regards,

Author Response

Dear Reviewer,

Thank you very much for your valuable comments.

We have addressed them in detail in our responses.

Kind regards,

Alexander Gegov

Round 3

Reviewer 2 Report

Dear Authors,

Thank you for your response letter. Regarding the 3rd round review of the manuscript, it can be accepted for further processing.

Regards,

This manuscript is a resubmission of an earlier submission. The following is a list of the peer review reports and author responses from that submission.

Round 1

Reviewer 1 Report

In this paper, the mix method combined auto regressive with exogenous in-put (ARX)-Laguerre and fuzzy proportional-integral-derivative (PID) observation system is presented for pipeline leak detection and estimation. The experimental cases and verification are seems sound and interesting. However the innovation point of this paper is weak and poor, the reviewer does not recommend this work publish in electronics, and some key weak points are as follows: (1)Why this mix method is presented, compared with existing works? The innovation point or new contents are not clear, which drawback or flaw is or are addressed in your method? (2)As you said: “the PID observer based on the ARX-Laguerre model is designed to improve leak estimation in the presence of uncertainties.”, so the PID method here is designed to improve …..or the PID method here is designed to detect or identify? (3)The references regarding fuzzy PID observer are not enough. (4)Many equations are duplicated from textbook or papers that published for many decades, there's no need to list them all.

Reviewer 2 Report

In this manuscript (ms), the authors present an auto regressive with exogenous input (ARX)-Laguerre fuzzy proportional-integral-derivative (PID) observation system in order to detect and estimate a leak in pipelines.

The problem treated is remarkably interesting and the results presented are potentially important. The main issue with the present version of the ms is that the presentation of the mathematics (due to notation) is extremely hard to follow to the point that I could not judge the scientific soundness. Hence, I suggest that the authors should go through the ms carefully and try to improve the presentation of Sections 2, 3, and 4. All the equations should be free from typographical errors and the quantities used should defined clearly one by one. Moreover, the authors should try to avoid special symbols like the Angstrom (Å) unit or rounded d-bar as parameters because they unnecessarily complicate the ms.

Below I mention a few of these issues that the authors should treat in their revised ms:

  1. 50, “has been provided” -> “ has provided”
  2. 62, define the acronyms ARX and PID in the main text also.
  3. 74, ρ is also used below for mass density
  4. 76, please avoid using the Angstrom (Å) unit for cross-sectional area
  5. Figure 1, change calligraphic H to calligraphic P because the H’s are defined much later
  6. Clearly define the quantities rounded d-bar, calligraphic F, and the two v’s of Equation (1) close to the equation.
  7. 84, please define these quantities earlier
  8. 91, please provide a reference
  9. Equation 9, define Π.
  10. l.114, z_{leak} should be s_{leak}
  11. Figure 2, “S”-> “s_{leak}”. Probably?
  12. As said, the notation in Section 3 is incomprehensible starting with two kinds of S (capital and lower case) in Equation (3.1) which is labeled (1)! There are subscripts like n which do not appear to be summated, capital X and lowercase x appear. Please define correctly and rigorously all the quantities of the Equation appearing between lines 137 and 138 on page 4. Please renumber this and all following Equations.
  13. The same applies to the Equation labeled (2) on page 5. What is S and how it can be transposed?
  14. Figure 3 should at least connect some of the quantities of Section 2 to those of Section 3 in order to be useful for the reader.
  15. Equation labeled (4) on page 5, please define all the quantities that appear in this Equation. Pipe inflow and outflow should be related to the quantities discussed in Section 2 for the benefit of the reader’s understanding.
  16. Please eliminate the typos in Equation labeled (8) on page 6.
  17. 165-168, do not use italics.
  18. 169, Please define clearly the quantities that appear in Equation labeled (10) on page 6, e.g., what is the state vector in term of the quantities introduced in Section 2?
  19. The rest of Section 4 should be carefully checked for typos and rigorous definitions of all quantities present.
  20. In Equation labeled (26) on page 10, please delete “.,”
  21. In reference [26], the correct publisher and address is Hemisphere Publishing Corp., Washington, D.C.; McGraw-Hill Book Co., New York.

Thus, the ms needs a major revision before I can suggest its publication in Electronics.

Reviewer 3 Report

Dear Authors,

Regarding the first-round review of this manuscript, the reviewer believes that the contribution of this manuscript is low, and the method and results of this manuscript are very similar to the following papers:

  1. Piltan, Farzin, and Jong-Myon Kim. "Pipeline Leak Detection and Estimation Using Fuzzy-Based PI Observer." In International Conference on Intelligent and Fuzzy Systems, pp. 1122-1129. Springer, Cham, 2019.
  2. Piltan, Farzin, and Jong-Myon Kim. "Advanced fuzzy-based leak detection and size estimation for pipelines." Journal of Intelligent & Fuzzy Systems 38, no. 1 (2020): 947-961.
  3. Piltan, Farzin, Muhammad Sohaib, and Jong-Myon Kim. "Fault diagnosis of a robot manipulator based on an ARX-Laguerre fuzzy PID observer." In International Conference on Robot Intelligence Technology and Applications, pp. 393-407. Springer, Cham, 2017.

So, because of these reasons and more, the reviewer suggests rejecting this manuscript.

Regards,

Reviewer 4 Report

The paper presents a very interesting topic. However, the quality of the current version of the paper is generally between average and low, thus I cannot recommend the publication, at least in the present form.

Follows a list of comments to help the authors to improve the paper quality.

In section 1 (Introduction) related works are presented. However, it would have been helpful to provide some comparative information with the proposed new estimation method based on the fuzzy PID observer.

In addition, what is the advantage(s) of this new proposed method is not specified.

The last paragraph, which describes the remainder of the paper needs some elaboration: the titles of the specific sections must be specified.

In equation (1) the meaning for some of the parameters is not specified.

The elements of the methodology presented in Figure 3 are not all of them explained sufficiently.

The text on lines 166-168 needs not to be in italic.

The section 4 (ARX-Laguerre Fuzzy PID Observation Technique) and particularly Section 4.2 (Fault diagnosis) needs further elaboration and explanations to be provided. The methodology of faults diagnosis is not presented sufficiently comprehensive, neither is very clear what is aimed to show with cases 1 to 3. Overall the proposed observation technique is not well presented and explained.

The specific values of the errors illustrated in Figures 6 and 7, and Figures 10 and 11, are not stated explicitly. Perhaps values representation in tables could assist in clarification of the specific error values obtained.

In section 5 (Simulation results) it is stated that the proposed method detects faults in less time in comparison with PID observer technique (illustrated in Figure 12). However once again there are no specific values of time measurements stated explicitly, neither a table of comparative representation of these values.

In addition, the less time required for the estimation of potential leakage fault seems to be the only advantage of this new proposed method examined in simulation only. However, it is not clear how much is less that value compared to other methods/techniques. From the Figure 12 it does not seem to be considerable. If this is the case, then it seems that there is no significant contribution of the proposed new method.